# Risk Assessment and Limiting Soil Factors for Vine Production—Cu and Zn Contents in Vineyard Soils in Galicia (Rías Baixas D.O.)

Raquel Vázquez-Blanco, Rocío González-Feijoo, Claudia Campillo-Cora, David Fernández-Calviño 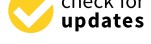 and Daniel Arenas-Lago *

Departamento de Bioloxía Vexetal e Ciencia do Solo, Área de Edafoloxía e Química Agrícola, Facultade de Ciencias, Universidade de Vigo, As Lagoas s/n, 32004 Ourense, Spain
* Correspondence: darenas@uvigo.es

**Abstract:** Characteristics of vineyard soils and management practices can be assessed to determine the soil trend evolution, risks, and limits of soils for vine production through soil factors and foliar diagnosis. This study was made with soils from a vineyard divided into two plots belonging to the Rías Baixas D.O. The vineyard soils were sampled and characterized for three years. The total and available Cu and Zn contents and the physicochemical characteristics of the soils were determined annually and every four months, respectively. The main objective was to assess edaphic properties, phytosanitary treatments, fertilization, and tillage applied to indicate the quality of the vineyard soils. The soils presented certain limitations associated with mechanization, trafficability, and ease of tillage for cultivation. The soils showed a sandy loam texture, which makes the application of compost necessary to improve water retention and cation exchange capacity. Phytosanitary treatments and fungicides caused phytotoxic contents of Cu and Zn in the soils without being detrimental to the vines. In conclusion, the edaphic factors and foliar analysis were adequate to evaluate the condition of the soils and vines and to establish the necessary measures to improve the edaphic conditions of the vineyard soils to improve plant production.

**Keywords:** copper; zinc; fungicide; phytosanitary; foliar diagnostic; nutritional status

## 1. Introduction

In recent decades, the wine sector in Galicia (Spain) has acquired great importance, especially the Rías Baixas Denomination of Origin (D.O.). According to data from the Rías Baixas D.O., grape and wine production has increased considerably ($\approx 600\%$) in the last 30 years along with exports and sales [1]. Moreover, the hectares dedicated to vine monoculture have increased considerably (currently almost 5000) since many soils devoted to other activities are being converted into vineyards to increase the production of this crop in this area [2]. All this, in the context of growing winemaking activity in the Rías Baixas D.O., has increased the extensive and intensive use of phytosanitary treatments, mainly fungicides based in Cu and other elements, such as Zn, to improve the quality and quantity of grape and wine production. Historically, Cu-based and Zn-based fungicides have been used for more than a century and a half to treat different grapevine diseases caused by powdery mildew [*Uncinula necátor* (Schwein.) Burrill)], mildew [*Plasmopara vitícola* (Berk. et Curtis ex. de Bary) Berl. et de Toni], and phylloxera [*Daktulosphaira vitifoliae* Fitch]. Despite the necessary and beneficial use of fungicides in vineyards, their prolonged and uncontrolled use has led to an accumulation of Cu and also Zn in many vineyard soils. These elements have led to soil degradation and severe environmental problems in other ecosystem compartments [3,4].

Copper generally shows low mobility in soils and tends to accumulate in the first few centimeters due to its tendency to form organometallic complexes with organic matter. In

any case, its sorption on carbonates, clay minerals, and Mn and Fe oxyhydroxides makes it possible that Cu can be found in deeper soil horizons. Zinc is an essential micronutrient for plants with high mobility and bioavailability [5]. This element may be a free ion $Zn^{2+}$ in soils, showing a high solubility at an acidic pH. Zinc is also fixed to organic matter through complexation or chelation processes and to clay through surface adsorption processes, all of which are pH-dependent [6]. Although Cu and Zn are essential elements for living beings, an excess of them may cause toxicity problems in plants or other organisms. The most common symptoms of Cu toxicity observed in plants are leaf darkening, chlorosis, and root malformation [7,8]. Among plants' most common Zn toxicity symptoms are growth problems, biomass loss, chlorosis, photosynthesis inhibition, nutrient imbalance, root darkening, cell damage as lipid peroxidation, and membrane damage [9–11]. Copper and Zn from fungicides are generally introduced into the soil by direct deposition, washing of foliage, or falling leaves. In the last decades, several studies pointed out that vineyard soils with high Cu contents need remediation strategies since Cu may cause a high toxicity risk in vines [12–18]. Likewise, wild plants growing spontaneously in vineyard soils or grasses used as cover crops are also affected by excessive Cu accumulation in vineyard soils. In general, the application of Cu-based fungicides also affects the retention of herbicides and/or insecticides, which may negatively influence their adsorption process and cause a greater risk of leaching these compounds [19–23].

In addition, the topographical characteristics of the Rías Baixas D.O. area (mild–steep slopes) and the soil characteristics (sandy loam texture, acid pH, shallow depth, organic matter high contents, and lack of nutrients because of high rainfall washing) favor the accumulation of Cu in vineyard soils. The Atlantic climatic conditions in Rias Baixas, continuous rainfall (>1400 mm $year^{-1}$), mild temperatures, and high humidity, favor the spread of pests [24], which makes it necessary to increase the use of fungicides in vineyards. In addition, different management practices (soil plowing, application of background fertilizer, compost, herbicides, or pruning of vines) may alter the edaphic characteristics of vineyard soils and increase Cu and Zn availability and mobility. All this causes environmental issues in other compartments of the ecosystems (e.g., hydrosphere, other soils, crops, and/or biota) [25–27].

Vineyard soil alterations can be evaluated using soil factors that accurately determine the situation of vineyard soils and their evolutionary trends. This is helpful to establish adequate soil management measures and maintain levels of fertility, ensuring soil productivity and environmental quality [28,29]. Among these soil factors, it is necessary to highlight those related to soil characteristics (e.g., effective depth, stoniness, erosion risk associated with the slope, organic matter content, N and P contents, C/N ratio, effective cation exchange capacity, macronutrient (K, Mg, Ca) balance, pH, and/or salinity) [28,30]. Moreover, total metal contents, particularly Cu and Zn, are used as indicators to assess vineyard soil pollution associated with fungicides [4,31,32]. In any case, the total metal content is not usually a good indicator to assess pollution in vineyard soils. Several studies have shown that to study the contamination and the possible toxicity, mobility, distribution, and/or transport of metals in vineyard soils, it is much better to determine the (bio)available metal content [4,33–35].

On the other hand, plant analysis is the most appropriate method to determine the nutritional status of a vineyard [36,37] and how different soil factors can affect vine production. Various studies have indicated that nutrient concentration in mature leaves of woody species can provide adequate information on the overall nutritional status of vines. Leaf diagnosis is based on the fact that the leaf is the most metabolically active organ in the plant, and nutritional alterations affect it more than other organs [36].

This study goes deep on the fact that the continuous cultivation of *Vitis vinifera* L., the management of soil practices, and intensive phytosanitary treatments may modify the quality and fertility of vineyard soils. These modifications can be evaluated through soil factors that, once related, may serve to assess the quality of the vineyard soils and their evolutionary trends. This will make it possible to establish corrective measures to

maintain adequate fertility and environmental quality and increase vineyard productivity. The increasing number of hectares dedicated to vine monoculture in the Rías Baixas D.O. makes it necessary to know the quality of the soils on which they are grown as well as the limiting factors for production.

Therefore, the main objective of this study is to assess and identify edaphic properties indicative of the quality of vineyard soils, determine their critical thresholds, and develop a soil health monitoring network through soil factors for vineyards soils in the Rias Baixas D.O. The specific objectives are: (i) to identify suitable vineyard soil properties, including establishing critical thresholds for them and assessing soil quality, (ii) to determine and assess the total and available contents of Cu and Zn in vineyard soils to know whether there are pollution problems derivate from using fungicides, (iii) to know whether alterations of the characteristics of vineyard soil derived from fungicide application and other management practices, such as the application of an organic amendment, can be evaluated through soil factors that provide information about the quality and evolutionary trend of vineyard soils; and (iv) to assess the nutritional status of vines in the vineyards studied to know how the different soil factors affect plant production.

## 2. Materials and Methods

### 2.1. Study Area

This study was made with soils from a vineyard (3.5 ha—30 years old) belonging to the Rías Baixas D.O. located in O Salnés region in the municipality of Vilanova de Arousa (Galicia, Spain) (Figure 1). The vineyard consists of two plots separated by a stream. Plot A is practically flat (slope < 1%). Plot B has a slope of approximately 7% with an intermediate zone that is almost flat (slope < 1%). In both, *V. vinifera* (Albariño grape variety) is grown for vinification purposes.

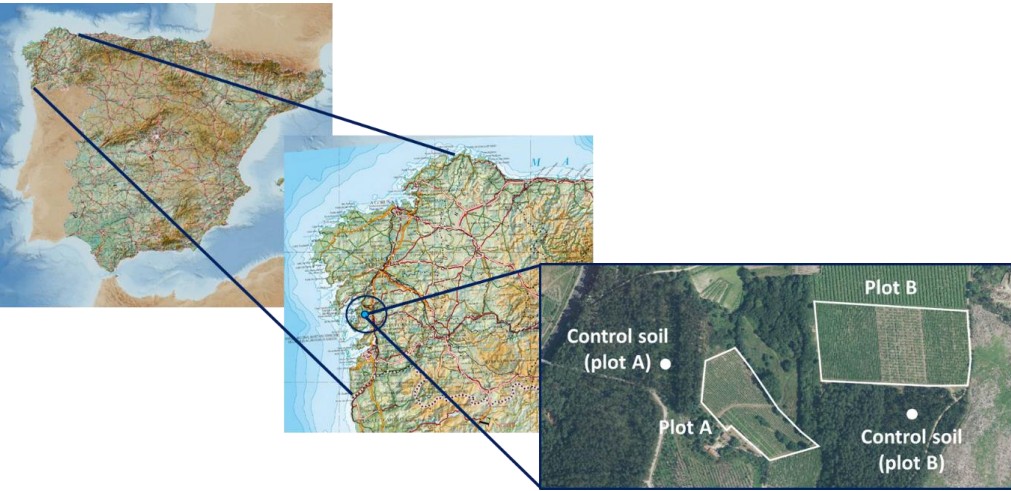

**Figure 1.** Location of the sampled vineyard plots and control soils.

The control soil next to plot (A) is located in a eucalyptus forest of *Eucalyptus globulus* Labill that also grows *Pteridium aquilinum* (L.) Kuhn. The control soil next to plot (B) is located in a forest area where there is tree vegetation, such as *Quercus suber* L., *Salix atrocinerea* Brot, *Corylus avellana* L., and *Rubus ulmifolius* Schott, and diverse herbaceous plants, such as *Festuca* sp., *Lolium* sp., *Agrostis* sp., and *Trifolium* sp.

The soil selected from plots A and B were classified as Mollic Umbrisols, while both controls were classified as Cambic Umbrisols [38].

### 2.2. Soil Sample Collection

Soil sampling was carried out in the two plots (A and B) by taking the samples from the topsoil (0–30 cm) at different points. In plots A and B, six and eight points were sampled, respectively. At each sampling point, three soil samples were taken with an Eijkelkam

sampler (Royal Eijkelkamp, Giesbeek, The Netherlands) making a compound sample stored in polythene bags. In the laboratory, the soil samples from each sampling point were air-dried and sieved through 2 mm mesh, homogenized in a vibratory homogenizer for solid samples (Fritsch rotary sampler divider Laborette 27, Fritsch GmbH, Idar-Oberstein, Germany), and stored for later analysis. In addition, two control soils were sampled in adjacent areas of each plot following the same sampling procedure. All soils, including controls, were sampled every four months (October, March, and July) for three years.

### 2.3. Vineyard Maintenance

During the sampling period, the vineyard did not undergo mechanization processes, so the soil texture, structure, and stoniness were not mechanically altered during the three years of sampling. Spontaneous herbaceous vegetation was maintained under the vines cut and harvested after the grape harvest. This vegetation, together with that collected under the control areas' deciduous forest, was used to obtain compost during the first and second years in October. This compost was stored and added to the vineyard soils in the early spring before the second and fifth samplings without having been added before this study. The most relevant characteristics of this compost are shown in Supplementary Materials (Supplementary Materials Table S1). Moreover, the phytosanitary treatments used during the studied period are shown in Supplementary Materials (Supplementary Materials Table S2).

### 2.4. Soil Analysis

The most relevant physicochemical characteristics were analyzed to evaluate the limiting factors for the correct development of the vine, performing four-month analyses (October, March, and July) for three years according to the following methodology:

Particle-size analysis and soil texture were determined [39]. Soil pH was determined with an electrode (Crison basic 20, Hach Company, Ames, IA, USA) in 2:1 water-to-soil extract. Soil stoniness was determined following standard procedures [40], and particle density and bulk density were also measured [41]. Organic carbon content was determined by the Walkley and Black method [42]. The organic matter content was calculated by multiplying the total carbon content by the Van Bemmelen factor (1.724 for carbon contents lower than 5.8 g kg$^{-1}$ and by 2.0 if higher). Total Kjeldahl-N (organic plus ammonium-N) was determined [43]. The P content was extracted using the Mehlich-3 method [44]. The analyses were performed by inductively coupled plasma optical emission spectroscopy (ICP-OES) (Perkin Elmer Optima 4300DV, PerkinElmer Inc., Waltham, Massachusetts, USA). The effective cation exchange capacity (ECEC) of the soil and exchangeable cations were determined following the methodology of Hendershot and Duquette [45].

The available Cu and Zn contents were extracted with 20 mL of extraction solution DTPA (0.005 M DTPA, 0.01 M CaCl$_2$, and 0.1 M TEA adjusted to pH 7.3) in a relation 1/10 (soil/solution) after shaking for 2 h [46]. The pseudototal (total hereinafter) Cu and Zn content were extracted using acid digestion with aqua regia (HNO$_3$:HCl; ratio 1:1) in Teflon containers placed in a microwave oven (Ethos 1; Milestone Srl, Sorisole, Italy). Cu and Zn analysis in the extracts was carried out by ICP-OES (Perkin Elmer Optima 4300DV). The Cu and Zn contents in the vineyard soils were analyzed annually (once per year) at the beginning of October after the grape harvest.

### 2.5. Limiting Factors for Vine Development

A series of physicochemical parameters were defined to evaluate the quality of the soil for the correct development of the vine by assigning critical values based on the Soil Fertility Capability Classification (SFCC) model [47] and adapted for Galicia [48–50] with some modifications for vineyard soils.

### 2.5.1. Physical Factors

- Texture of the arable layer. Textural classes S, L, C, and O are established (and R when there is a lithic contact below the arable layer). Their ranges are S—Sandy (sandy and sandy loams); L—Loamy <35% clay (excluding sandy and sandy loams); C—Clay >35% clay; O—Organic >30% organic matter up to 50 cm or more.
- Stoniness. Two stoniness intervals are defined: medium stoniness (15–35%) and stony (>35%) for a fraction greater than 2 mm.
- "A" factor. This indicator evaluates the presence of rocky outcrops. Soils with exposed rocks between 10% and 90% are considered limiting for cultivation.
- "E" factor. This indicator evaluates the risk of erosion based on the slope of the land and other parameters such as the erosive potential of rainfall, soil erosivity, conservation practices, and vegetation cover. The slope is classified into areas of strong (>25%), medium (6% and 25%), and low (<6%).
- "M" factor. This indicator evaluates the impediments to mechanization due to trafficability and ease of tillage. It is classified into four levels of limitation using the slope as the basic criterion: steep (>15%), medium (7–15%), slight (3–7%), and none (<3%).

### 2.5.2. Chemical Factors

- Acidity factor. This indicator evaluates the pH ranges of soils, which are established for vineyard soils according to strongly acidic (3.5 and 4.5), acidic (4.5 and 5.5), weakly acidic (5.5–6.5), neutral (6.5–7.5), and basic (>7.5). Likewise, this indicator also allows for assessing the percentage of Al saturation in the exchange complex. Its presence is considered moderate between 10 and 20%, strong between 20 and 60%, and very strong higher than 60% (alic character).
- "a" factor. This indicator evaluates the alic character and is applied in soils with high Al concentrations when its saturation percentage in the exchange complex is higher than 60%.
- "om" factor. This is used to evaluate the organic matter content. It is considered deficit contents < 2% (om1) and excess contents >5% (om2).
- "N" factor. This indicator evaluates the nitrogen content. It is applied to soil whose total N content is <0.1% (low) or >0.3% (high).
- "P" factor. This indicator evaluates the phosphorus content and delimits excessive values for viticulture crops if they imply unnecessary inputs: low < 18 $mg\ kg^{-1}$, medium 18–36 $mg\ kg^{-1}$, high 36–72 $mg\ kg^{-1}$, very high > 72 $mg\ kg^{-1}$.
- "e" factor. This indicator evaluates the low cation exchange capacity. There is a strong limitation when ECEC is < 4 $cmol_{(+)}\ kg^{-1}$ and moderate if it is between 4 and 7 $cmol_{(+)}\ kg^{-1}$.
- "Ca" factor. This indicator evaluates Ca deficiency. It is considered deficient when the value of the exchange complex is <1.5 $cmol_{(+)}\ kg^{-1}$ or the Ca/Mg ratio < 0.5.
- "n" factor. This indicator evaluates the presence of high Na contents in the exchange complex. It is considered positive when Na saturation in the exchange complex is ≥15%.
- *"k1" factor*. This indicator evaluates the low reserve of K (< 0.2 $cmol_{(+)}\ kg^{-1}$).
- *"k2" factor*. This indicator evaluates the unbalance in the exchange complex between K and the rest of the cations (<2% to the sum of the exchange bases when <10 $cmol_{(+)}\ kg^{-1}$).
- *"Mg" factor*. This indicator evaluates Mg deficiency. It is applied in soils with insufficient assimilable Mg content. This deficit may be due to a low content (<0.4 $cmol_{(+)}\ kg^{-1}$) or a Ca/Mg imbalance (>10 imbalance) or a K/Mg imbalance (>0.5).
- *"Cu" factor*. This indicator evaluates the Cu content. Available Cu values <1 $mg\ kg^{-1}$ (Cu1) are considered deficient. Available contents higher than 25 $mg\ kg^{-1}$ (Cu2) in sandy soils can be toxic for vines, mainly in soils with acidic pH. Generally, total contents above 100 $mg\ kg^{-1}$ (Cu3) are considered phytotoxic [5].

- "Zn" factor. This indicator evaluates the Zn content. Available Zn contents <3.3 mg kg$^{-1}$ (Zn1) are considered deficient for viticulture crops, and >15 mg kg$^{-1}$ (Zn2) are phytotoxic. In general, upper toxic levels for total contents of Zn are considered 500 mg kg$^{-1}$ (Zn3) [5].

### 2.6. Determination of Vine Nutritional State

To determine vine nutritional state, a foliar diagnosis was carried out by sampling vine leaves (petiole and leaf blade) at veraison during the second and third year of the study. For this purpose, leaves were collected from different vines in each vineyard (plots A and B). Border vines or vines with characteristics very different from the population mean were avoided. Leaves were selected opposite the basal cluster of a fruiting shoot, which is considered the most representative of plant nutritional status. A representative number of leaves were collected, taking two per vine and separating the leaf blades from the petioles. The leaf samples taken in each zone were used to prepare composite samples (leaf blades and petioles), from which, three subsamples were taken and analyzed in triplicate.

Based on the nutritional levels provided by various bibliographic sources [51–54], the nutritional status of the vines was analyzed according to Table 1:

**Table 1.** Guide to nutrient deficiencies in vine leaves.

| Status | Leaf Blade | | | | | | | | | Petiole | |
|---|---|---|---|---|---|---|---|---|---|---|---|
| | N | P | K | Zn | Cu | Mg | Ca | Fe | Mn | Status | K/Mg |
| | | | | | mg kg$^{-1}$ | | | | | | |
| Deficiency | - | <1.5 | <10 | - | <4 | <2.0 | <20 | - | - | Probable K deficiency | <2 |
| Slight deficiency | <24 | 1.5–2.0 | 12 October | 0–30 | 5 April | 2.0–2.3 | 20–25 | 0–50 | - | Risk of K deficiency | 3 |
| Optimum | 24–26 | 2.0–2.4 | 14 December | 30–150 | 20 May | 2.3–2.7 | >25 | 50–225 | 0–200 | Normal feeding | 10 |
| Slight excess | 26–28 | 2.4–2.6 | 14–16 | 150–400 | 20–40 | 2.7–3.0 | - | 225–300 | 200–500 | Risk of Mg deficiency | 12 |
| Excess | >28 | >2.6 | >16 | >400 | >40 | >3.0 | - | >300 | >500 | Probable Mg deficiency | >12 |

In addition, for the analysis of macronutrients in the petioles, N contents >6 mg kg$^{-1}$ correspond to normal nitrogen nutrition and P contents >1.5 mg kg$^{-1}$ to normal phosphorus nutrition.

### Leaves Analysis

For leaf analysis, leaf blades and petioles were separated and washed first with water and then with distilled water. The samples were dried with a centrifuge (ROTINA 380, Hettich, Westphalia, Germany) placed in trays covered with filter paper, and dried in a forced air oven at 50 °C for 72 h. They were crushed in a blade grinder (Kinematica™ Polymix™ PX-MFC 90 D, Thermo Fisher Scientific, Waltham, MA, USA), ground with an agate mill (RM 200 Retsch, Verder Scientific GmbH & Co. KG, Haan, Germany), and sieved with 0.5 mm mesh. Total Kjeldahl-N (organic plus ammonium-N) was determined [43]. The contents of K, Ca, Mg, P, Fe, Mn, Cu, and Zn were determined by ICP-OES (Perkin Elmer Optima 4300DV) in the extracts obtained after digestion of the samples with H$_2$SO$_4$ (95%) and H$_2$O$_2$ (30%) [55].

### 2.7. Statistical Analysis

Soil analysis was made in triplicate at each soil sample (plot A: $n = 6$; plot B: $n = 8$; controls: $n = 1$) area. The results presented are the maximum, minimum, and mean values obtained for each vineyard standard deviation of the mean for all soil samples from each vineyard plot. ANOVA and Duncan's multiple range tests were used to compare differences between the different soil samples (See Supplementary Materials). Fisher's minimum

significant difference (LSD) test at 5% was used to compare means with weighted variance. The Kolmogorov–Smirnov test and Levene's test were applied to verify the data's normality and the variances' homogeneity in order to test their homoscedasticity, respectively. The detection limits of the ICP-OES (Perkin Elmer Optima 4300DV) are 0.01 mg $L^{-1}$ for Cu, Zn, Al, Mn, and Mg; 0.02 mg $L^{-1}$ for Ca; and 0.1 mg $L^{-1}$ for Fe, P, and Na. The detection limit for N is 1.0 mg $L^{-1}$.

## 3. Results

### 3.1. Analysis of the Limiting Factors for Vine Development Physical Properties of Soilsubsection

The physical properties of the soils were analyzed only for the soil samples taken in the first sampling since physical properties, such as bulk density, stoniness, and texture, are not very susceptible to variation over time. The physical characteristics of the soils are shown in Table 2 (full data in Supplementary Materials Table S3). All soils from both plots have a sandy loam texture, which implies permeability, specific surface area, compactness, nutrient storage capacity, water storage capacity, and average water retention. In contrast, the control soils have a loamy-sand texture with a slightly higher percentage of clay than the vineyard soils. The stoniness of soils from plot A varies between 52 and 62%. All plot A soils are stony soils (>35%). For soils from plot B, the stoniness varies between 14% (slightly stony and 63% (stony)). The bulk density of the vineyard soils varies between 0.9 and 1.3 g·cm$^{-3}$, being much lower (0.3–0.4 g cm$^{-3}$) in the controls since these soils are rich in organic matter and may contain friable clays, which gives them a low density and greater porosity.

**Table 2.** Physical properties of vineyard soils and controls.

| Soil | Bulk Density (g cm$^{-3}$) | Stoniness (%) | Sand (%) | Silt (%) | Clay (%) | Texture |
|------|------|------|------|------|------|------|
| $A_{min}$ | 0.96 | 52.01 | 78.7 | 16.1 | 3.4 | Sandy-Loam |
| $A_{max}$ | 1.35 | 61.54 | 79.9 | 16.7 | 5.2 | Sandy-Loam |
| $A_{(mean)}$ | 1.15 ± 0.19 | 57.53 ± 3.96 | 79.3 ± 0.7 | 16.4 ± 0.3 | 4.3 ± 1.0 | Sandy-Loam |
| $B_{min}$ | 0.98 | 14.47 | 75.9 | 17.6 | 6.2 | Sandy-Loam |
| $B_{max}$ | 1.14 | 63.87 | 76.1 | 18.1 | 6.4 | Sandy-Loam |
| $B_{(mean)}$ | 1.04 ± 0.07 | 38.32 ± 14.50 | 75.9 ± 0.3 | 17.8 ± 0.2 | 6.3 ± 0.1 | Sandy-Loam |
| CA | 0.35 ± 0.04 | 44.62 ± 8.32 | 71.8 ± 0.2 | 19.6 ± 0.1 | 8.6 ± 0.0 | Loamy-Sand |
| CB | 0.44 ± 0.05 | 10.24 ± 2.45 | 75.5 ± 0.1 | 15.8 ± 0.1 | 8.7 ± 0.0 | Loamy-Sand |

max: maximum value; min: minimum value; mean: average ± standard deviation.

The limiting factors for vine development related to the physical characteristics of the soils indicate:

- Texture of the arable layer. The vineyard soils of both plots have a texture class of sandy loam with less than 30% of organic matter, which is classified as S (Sandy), while soil controls are classified as L (Loamy).
- Stoniness and "A" factor. In general, the vineyard soils of both plots are classified as stony (>35%) except for soil B5 (24.4%), which is medium stoniness (15–35%), and soil B4 (14.5%), which is not limited by the presence of stoniness (<15%). On the other hand, no rocky outcrops were found in the vineyard soils studied, so the soils did not present a limitation associated with factor "A" for the cultivation of grapevines.
- "E" factor and "M" factor. According to the slope, plot B has a slope of ≈7%, while the slope in plot A is <1%. Thus, the risk of soil erosion in both plots is considered low. Regarding the impediments associated with mechanization, trafficability, and easy tillage, the vineyard soils from plot B are regarded as "medium/slight limitation" for these management practices. In contrast, the vineyard soils from plot A are considered "with no limitation."

*3.2. Analysis of the Limiting Factors for Vine Development: Chemical Properties of Soils*

The data of chemical soil characteristics analyzed for three years—including three annual sampling campaigns (October, March, and July)—are shown in Table 3 (full data in Supplementary Materials Tables S4–S6).

The limiting factors for vine development related to the chemical characteristics were evaluated over three years (and three samplings per year) to establish the evolution of the two vineyards during the studied period and their comparison with the respective controls.

- Acidity factor. The analysis of the pH of the vineyard soils during the nine samplings carried out shows that the soils of plot A and plot B have an average neutral pH (6.5–7.5), except for the pH of soils measured during the first sampling in the plot A and plot B, which are mostly weakly acidic (5.5–6.5). This is related to the fact that the first sampling of the study was done before adding compost (Supplementary Materials Table S1), whose pH of 8.1 favors the increase of soil pH. This indicates that the addition of compost was a sufficient measure to increase the pH of the soils and maintain correct pH values for the development of the vines. The percentage of Al saturation in the exchange complex in the soils of plot A and plot B is mostly < 1%, which explains why the potential pH $_{(KCl)}$ of the vineyard soils is not drastically much lower than the actual pH $_{(H_2O)}$. On the other hand, the control soils have percentages of Al in the exchange complex that are considered strong (20–60%) and very strong (>60), with values of 46% (plot A control) and 65% (plot B control). The control soil for plot B shows an alkali character (*"a" factor*).

- "om" factor. The percentages of organic matter in the vineyard soils determined during the first sampling are much lower than those found in the rest of the samples. This is due to the application of compost with an organic matter content of 34.9% (first application and 33.9% (second application)). On the other hand, the percentage of organic matter in the vineyard soils is also lower than in the control soils (forest soils), which indicates the loss of organic matter when the land use was changed from forest to vineyard. This confirms the need to apply external inputs (in this case, compost) to maintain the organic matter content in the vineyard soils. In general, the contents of organic material in plot A and plot B in the first sampling are within the soils considered to have normal levels of organic material (2–5%). Despite this, some of the soils of plot A have deficit contents (<2% - om1). However, from the second sampling onwards, the average percentages of organic matter increased and contained excess contents (>5% - om2).

- "N" factor. The total N content in the soils from both plots during the first sampling have low total N contents (<0.1%). After compost application (total N; 1st treatment: 25.2 g kg$^{-1}$ and 2nd treatment: 27.2 g kg$^{-1}$), the total N contents in the soils from plot A were normal. The only total N contents that exceeded 0.3% (considered high) were during the second and fifth samplings (March of the first and second year), just after compost application. In plot B, the total N contents are considered normal but may be regarded as high just after compost application. The control soils present excess N with total contents higher than 0.3%, and the plot A control is excessively high (>1%).

- "P" factor. The average P contents in vineyard soils ranged from high (36–72 mg kg$^{-1}$) to very high (high > 72 mg kg$^{-1}$) during the whole period studied. In any case, some of the vineyard soils have medium P contents (18–36 mg kg$^{-1}$). Therefore, there is no P deficit in any of the vineyard soils over the three years of the study. In contrast, the control soils have low phosphorus contents (<18 mg kg$^{-1}$), which indicates that the nearby forest soils are deficient in P and that any transformation of these lands for use as vineyards will require inputs of this macronutrient.

- "e" factor. The mean values of eCEC in the vineyard soils are all greater than 7 cmol$_{(+)}$ kg$^{-1}$, so none of the soils show limitations associated with eCEC. It should be noted that before applying the compost, the eCEC for some vineyard soils from plot B showed a moderate limitation according to this factor, with eCEC between 4 and 7 cmol$_{(+)}$ kg$^{-1}$.

- "Ca" factor, "n" factor, "k1" factor, "k2" factor, and "Mg" factor. The factors associated with the basic cations of the exchange complex indicate that none of the vineyard soils presented $Ca^{2+}$ deficiencies during the three years in which the sampling was carried out. On the other hand, the control soil of plot A showed an exchangeable Ca deficiency, with lower $Ca^{2+}$ values lower than 1.5 $cmol_{(+)}$ $kg^{-1}$. Regarding the $Na^+$ content in the exchange complex, none of the vineyard soils or the controls had excess $Na^+$ ($\geq 15\%$) in the exchange complex. Vineyard soils also do not have low K reserves (<0.2 $cmol_{(+)}$ $kg^{-1}$) nor an imbalance of the exchange complex (neither k1 nor k2). An imbalance between Ca and Mg content (ratio >10) was observed in some vineyard soils, mainly after compost application before the second and third sampling, with average ratio values of 13.2 and 13.7, respectively. Subsequently, a progressive decrease in this ratio was observed until values below 10. In addition, several soils in both plots show an imbalance between Mg and K (ratio > 0.5). The control soil of plot A also has an Mg unbalance, while the ratio of the control soil of plot B is within the limit established for Mg content to be considered unbalanced.

**Table 3.** Chemical soil characteristics.

| Soil | pH$_{(H_2O)}$ | pH$_{(KCl)}$ | Org C | OM | N | K | P | C/N | K⁺ | Ca²⁺ | Mg²⁺ | Al³⁺ | Na⁺ | eCEC | Ca/Mg | K/Mg |
|---|---|---|---|---|---|---|---|---|---|---|---|---|---|---|---|---|
| | | | g kg⁻¹ | % | g kg⁻¹ | mg kg⁻¹ | | | cmol$_{(+)}$ kg⁻¹ | | | | | | | |
| **1st sampling—October (Year 1)** | | | | | | | | | | | | | | | | |
| **A$_{min}$** | 5.6 | 4.7 | 7.8 | 1.4 | 0.6 | 248 | 67 | 11.1 | 0.6 | 4.4 | 2.0 | 0.1 | 0.1 | 7.4 | 2.2 | 0.2 |
| **A$_{max}$** | 6.1 | 5.8 | 18.0 | 3.1 | 0.8 | 286 | 91 | 28.3 | 0.7 | 12.2 | 2.9 | 0.1 | 0.2 | 15.5 | 5.0 | 0.4 |
| **A$_{mean}$** | 5.8 | 5.4 | 13.1 | 2.3 | 0.7 | 270 | 80 | 19.1 | 0.7 | 8.3 | 2.5 | 0.1 | 0.1 | 11.6 | 3.3 | 0.3 |
| **SD** | 0.2 | 0.4 | 4.9 | 0.8 | 0.1 | 16 | 10 | 7.1 | 0.0 | 2.6 | 0.3 | 0.0 | 0.0 | 2.8 | 1.0 | 0.05 |
| **B$_{min}$** | 5.4 | 5.0 | 15.0 | 2.6 | 0.2 | 255 | 54 | 38.8 | 0.7 | 2.5 | 0.6 | 0.1 | 0.1 | 4.8 | 1.3 | 0.4 |
| **B$_{max}$** | 6.6 | 6.1 | 21.5 | 4.3 | 0.4 | 396 | 91 | 108 | 1.0 | 11.8 | 2.1 | 0.6 | 0.2 | 14.1 | 8.7 | 1.4 |
| **B$_{mean}$** | 6.2 | 5.5 | 18.1 | 3.4 | 0.3 | 324 | 82 | 69.0 | 0.8 | 4.4 | 1.5 | 0.3 | 0.1 | 7.1 | 3.5 | 0.7 |
| **SD** | 0.4 | 0.4 | 2.6 | 0.7 | 0.1 | 60 | 14 | 25.6 | 0.2 | 3.2 | 0.6 | 0.2 | 0.0 | 3.1 | 2.7 | 0.4 |
| **CA** | 4.9 | 4.3 | 56.0 | 11.2 | 11.3 | 158 | 2 | 5.4 | 0.3 | 0.2 | 0.2 | 1.6 | 0.2 | 2.5 | 1.1 | 1.6 |
| **SD** | 0.1 | 0.1 | 2.1 | 1.1 | 1.2 | 12 | 0 | 1.1 | 0.1 | 0.1 | 0.1 | 0.2 | 0.1 | 0.6 | 0.1 | 0.1 |
| **CB** | 4.9 | 3.9 | 44.0 | 8.8 | 3.9 | 162 | 2 | 12.7 | 0.3 | 1.9 | 0.7 | 2.7 | 0.2 | 5.8 | 2.7 | 0.5 |
| **SD** | 0.1 | 0.1 | 1.9 | 1.3 | 0.5 | 9.0 | 0 | 1.2 | 0.1 | 0.1 | 0.1 | 0.1 | 0.1 | 0.5 | 0.1 | 0.1 |
| **2nd sampling—March (Year 1)** | | | | | | | | | | | | | | | | |
| **A$_{min}$** | 6.3 | 5.5 | 23.5 | 4.7 | 2.0 | 202 | 45 | 8.0 | 0.5 | 13.3 | 0.7 | 0.1 | 0.1 | 16.6 | 5.8 | 0.3 |
| **A$_{max}$** | 7.2 | 6.4 | 36.5 | 7.3 | 4.2 | 560 | 103 | 13.4 | 1.4 | 24.6 | 3.2 | 0.1 | 0.2 | 26.6 | 27.0 | 0.7 |
| **A$_{mean}$** | 6.9 | 6.1 | 31.4 | 6.3 | 3.1 | 347 | 68 | 10.7 | 0.9 | 19.0 | 1.9 | 0.1 | 0.1 | 22.0 | 13.2 | 0.5 |
| **SD** | 0.3 | 0.4 | 4.4 | 0.9 | 0.8 | 131 | 20 | 2.1 | 0.3 | 3.9 | 0.9 | 0.0 | 0.0 | 3.4 | 9.0 | 0.2 |
| **B$_{min}$** | 6.6 | 5.8 | 21.0 | 4.2 | 1.9 | 214 | 76 | 10.9 | 0.5 | 13.6 | 3.9 | 0.1 | 0.1 | 20.1 | 1.9 | 0.1 |
| **B$_{max}$** | 7.5 | 6.8 | 48.5 | 9.7 | 4.3 | 474 | 131 | 11.8 | 1.2 | 38.8 | 10.6 | 0.1 | 0.2 | 44.2 | 8.4 | 0.2 |
| **B$_{mean}$** | 7.1 | 6.2 | 36.1 | 7.2 | 3.2 | 322 | 101 | 11.3 | 0.8 | 20.5 | 5.5 | 0.1 | 0.1 | 27.0 | 4.1 | 0.2 |
| **SD** | 0.3 | 0.3 | 9.9 | 2.0 | 0.9 | 86 | 19 | 0.3 | 0.2 | 7.8 | 2.2 | 0.0 | 0.0 | 7.9 | 2.0 | 0.04 |
| **CA** | 5.0 | 4.2 | 61.0 | 12.0 | 11.4 | 112 | 3 | 5.4 | 0.4 | 0.2 | 0.2 | 1.6 | 0.2 | 2.6 | 1.0 | 2.0 |
| **SD** | 0.1 | 0.1 | 2.3 | 1.0 | 1.3 | 14 | 1 | 0.9 | 0.1 | 0.1 | 0.1 | 0.1 | 0.1 | 0.5 | 0.1 | 0.1 |
| **CB** | 4.9 | 3.9 | 48.0 | 9.6 | 3.8 | 138 | 7 | 12.6 | 0.3 | 2.0 | 0.7 | 2.8 | 0.1 | 5.9 | 2.9 | 0.4 |
| **SD** | 0.2 | 0.1 | 2.1 | 1.5 | 0.6 | 8 | 0 | 1.0 | 0.1 | 0.2 | 0.1 | 0.2 | 0.1 | 0.7 | 0.1 | 0.1 |
| **3rd sampling—July (Year 1)** | | | | | | | | | | | | | | | | |
| **A$_{min}$** | 6.5 | 5.9 | 21.5 | 4.3 | 1.8 | 210 | 40 | 10.9 | 0.5 | 10.9 | 1.2 | 0.1 | 0.1 | 13.8 | 5.7 | 0.4 |
| **A$_{max}$** | 7.3 | 6.7 | 31.0 | 6.2 | 2.7 | 672 | 60 | 14.6 | 1.7 | 24.0 | 2.6 | 0.1 | 0.2 | 27.1 | 13.7 | 0.9 |
| **A$_{mean}$** | 6.9 | 6.3 | 27.0 | 5.4 | 2.3 | 362 | 52 | 12.1 | 0.9 | 15.7 | 1.8 | 0.1 | 0.2 | 18.7 | 9.5 | 0.5 |
| **SD** | 0.3 | 0.3 | 3.3 | 0.7 | 0.4 | 163 | 7 | 1.4 | 0.4 | 4.8 | 0.5 | 0.0 | 0.0 | 4.7 | 4.0 | 0.2 |

**Table 3.** *Cont.*

| Soil | pH$_{(H_2O)}$ | pH$_{(KCl)}$ | Org C | OM | N | K | P | C/N | K$^+$ | Ca$^{2+}$ | Mg$^{2+}$ | Al$^{3+}$ | Na$^+$ | eCEC | Ca/Mg | K/Mg |
|---|---|---|---|---|---|---|---|---|---|---|---|---|---|---|---|---|
| | | | g kg$^{-1}$ | % | g kg$^{-1}$ | mg kg$^{-1}$ | | | | | | cmol$_{(+)}$ kg$^{-1}$ | | | | |
| B$_{min}$ | 7.0 | 6.3 | 13.5 | 2.7 | 1.0 | 304 | 60 | 12.6 | 0.8 | 11.0 | 2.3 | 0.1 | 0.1 | 14.4 | 2.0 | 0.1 |
| B$_{max}$ | 7.6 | 7.1 | 53.5 | 10.7 | 3.2 | 480 | 99 | 16.8 | 1.2 | 23.5 | 8.4 | 0.1 | 0.2 | 30.5 | 4.7 | 0.4 |
| B$_{mean}$ | 7.2 | 6.6 | 34.7 | 6.9 | 2.3 | 389 | 79 | 14.8 | 1.0 | 16.9 | 5.5 | 0.1 | 0.2 | 23.7 | 3.4 | 0.2 |
| SD | 0.2 | 0.3 | 14.6 | 2.9 | 0.8 | 64 | 14 | 1.7 | 0.2 | 4.0 | 2.1 | 0.0 | 0.0 | 5.2 | 1.1 | 0.1 |
| CA | 4.9 | 4.2 | 61.5 | 12.3 | 11.5 | 113 | 3 | 5.3 | 0.3 | 0.2 | 0.2 | 1.5 | 0.2 | 2.4 | 1.0 | 1.5 |
| SD | 0.1 | 0.1 | 2.3 | 0.5 | 1.3 | 11 | 0 | 1.8 | 0.2 | 0.1 | 0.1 | 0.2 | 0.1 | 0.7 | 1.0 | 2.0 |
| CB | 4.8 | 3.9 | 47.5 | 9.5 | 3.8 | 138 | 8 | 12.5 | 0.3 | 1.8 | 0.8 | 2.8 | 0.2 | 5.9 | 2.3 | 0.4 |
| SD | 0.2 | 0.1 | 1.8 | 0.4 | 0.6 | 8 | 0 | 3.0 | 0.1 | 0.1 | 0.2 | 0.1 | 0.1 | 0.6 | 0.5 | 0.5 |
| | | | | | | 4th sampling—October (Year 2) | | | | | | | | | | |
| A$_{min}$ | 6.7 | 5.8 | 24.5 | 4.9 | 1.7 | 188 | 21 | 11.3 | 0.5 | 14.6 | 1.1 | 0.1 | 0.1 | 16.4 | 5.5 | 0.2 |
| A$_{max}$ | 7.2 | 6.6 | 35.0 | 7.0 | 3.1 | 612 | 69 | 18.9 | 1.5 | 22.5 | 3.0 | 0.1 | 0.2 | 25.6 | 13.4 | 0.5 |
| A$_{mean}$ | 7.0 | 6.3 | 29.7 | 5.9 | 2.3 | 325 | 51 | 13.6 | 0.8 | 18.3 | 2.1 | 0.1 | 0.2 | 21.5 | 9.5 | 0.4 |
| SD | 0.2 | 0.3 | 4.1 | 0.8 | 0.6 | 152 | 17 | 3.0 | 0.4 | 3.2 | 0.7 | 0.0 | 0.0 | 3.5 | 3.3 | 0.1 |
| B$_{min}$ | 6.2 | 5.7 | 18.0 | 3.6 | 1.5 | 262 | 34 | 11.1 | 0.7 | 12.6 | 3.2 | 0.1 | 0.1 | 17.0 | 2.1 | 0.1 |
| B$_{max}$ | 7.6 | 7.0 | 49.0 | 9.8 | 3.2 | 424 | 127 | 18.5 | 1.1 | 25.7 | 6.4 | 0.1 | 0.2 | 29.9 | 8.1 | 0.3 |
| B$_{mean}$ | 7.1 | 6.4 | 32.7 | 6.5 | 2.3 | 351 | 69 | 14.1 | 0.9 | 16.8 | 4.6 | 0.1 | 0.2 | 22.5 | 4.1 | 0.2 |
| SD | 0.5 | 0.5 | 10.6 | 2.1 | 0.5 | 59 | 30 | 2.4 | 0.1 | 4.3 | 1.3 | 0.0 | 0.0 | 3.8 | 2.0 | 0.06 |
| CA | 4.8 | 4.2 | 56.1 | 11.2 | 11.5 | 110 | 2 | 4.9 | 0.4 | 0.2 | 0.3 | 1.6 | 0.2 | 2.7 | 0.7 | 1.3 |
| SD | 0.2 | 0.1 | 2.3 | 0.5 | 1.2 | 10 | 0 | 1.9 | 0.1 | 0.1 | 0.1 | 0.2 | 0.1 | 0.6 | 1.0 | 1.0 |
| CB | 4.8 | 4 | 43.1 | 8.6 | 4.0 | 135 | 9 | 10.8 | 0.3 | 1.9 | 0.7 | 2.7 | 0.2 | 5.8 | 2.7 | 0.4 |
| SD | 0.1 | 0.2 | 2.2 | 0.4 | 0.4 | 8 | 0 | 5.5 | 0.1 | 0.2 | 0.1 | 0.1 | 0.1 | 0.6 | 2.0 | 1.0 |
| | | | | | | 5th sampling—March (Year 2) | | | | | | | | | | |
| A$_{min}$ | 6.8 | 6.1 | 27.5 | 5.5 | 1.9 | 234 | 46 | 8.2 | 0.6 | 13.8 | 1.2 | 0.1 | 0.1 | 17.4 | 5.7 | 0.3 |
| A$_{max}$ | 7.4 | 6.8 | 34.0 | 6.8 | 4.0 | 570 | 83 | 15.3 | 1.4 | 21.4 | 3.2 | 0.1 | 0.2 | 23.7 | 16.7 | 0.7 |
| A$_{mean}$ | 7.1 | 6.5 | 30.9 | 6.2 | 3.1 | 356 | 66 | 10.4 | 0.9 | 17.2 | 2.0 | 0.1 | 0.2 | 20.4 | 10.2 | 0.5 |
| SD | 0.2 | 0.3 | 2.4 | 0.5 | 0.7 | 114 | 14 | 2.6 | 0.3 | 3.0 | 0.8 | 0.0 | 0.0 | 2.5 | 5.0 | 0.2 |
| B$_{min}$ | 7.0 | 6.2 | 19.0 | 3.8 | 1.2 | 254 | 37 | 11.6 | 0.6 | 12.8 | 2.7 | 0.1 | 0.1 | 18.5 | 2.1 | 0.1 |
| B$_{max}$ | 7.8 | 7.1 | 49.5 | 9.9 | 3.5 | 444 | 119 | 17.9 | 1.1 | 21.2 | 8.5 | 0.1 | 0.3 | 41.2 | 6.3 | 0.3 |
| B$_{mean}$ | 7.4 | 6.7 | 33.3 | 6.7 | 2.2 | 342 | 69 | 15.4 | 0.9 | 16.2 | 4.8 | 0.1 | 0.2 | 24.6 | 3.8 | 0.2 |
| SD | 0.3 | 0.3 | 9.9 | 2.0 | 0.8 | 54 | 28 | 2.4 | 0.1 | 2.7 | 1.8 | 0.0 | 0.1 | 7.5 | 1.4 | 0.06 |
| CA | 5.0 | 4.4 | 63.5 | 12.7 | 11.4 | 115 | 2 | 5.6 | 0.3 | 0.3 | 0.3 | 1.7 | 0.2 | 2.8 | 1.0 | 1.0 |
| SD | 0.3 | 0.1 | 2.5 | 0.5 | 1.2 | 12 | 0 | 2.1 | 0.1 | 0.1 | 0.1 | 0.2 | 0.1 | 0.6 | 1.0 | 1.0 |
| CB | 5.1 | 4.0 | 47.5 | 9.5 | 3.9 | 128 | 8 | 12.2 | 0.3 | 1.7 | 0.8 | 2.7 | 0.2 | 5.7 | 2.1 | 0.4 |
| SD | 0.1 | 0.2 | 1.8 | 0.4 | 0.4 | 8 | 0 | 4.5 | 0.2 | 0.3 | 0.1 | 0.1 | 0.1 | 0.8 | 3.0 | 2.0 |
| | | | | | | 6th sampling—July (Year 2) | | | | | | | | | | |
| A$_{min}$ | 6.5 | 5.7 | 25.5 | 5.1 | 1.9 | 244 | 50 | 8.7 | 0.6 | 7.7 | 1.2 | 0.1 | 0.1 | 11.1 | 2.3 | 0.2 |
| A$_{max}$ | 7.7 | 6.9 | 30.0 | 6.0 | 3.0 | 442 | 110 | 14.0 | 1.1 | 16.0 | 3.9 | 0.1 | 0.2 | 18.8 | 9.1 | 0.5 |
| A$_{mean}$ | 7.1 | 6.4 | 27.3 | 5.5 | 2.5 | 311 | 70 | 11.3 | 0.8 | 10.3 | 2.3 | 0.1 | 0.2 | 13.6 | 5.2 | 0.4 |
| SD | 0.5 | 0.5 | 1.8 | 0.4 | 0.4 | 73 | 23 | 2.1 | 0.2 | 3.0 | 0.9 | 0.0 | 0.0 | 2.7 | 3.0 | 0.2 |
| B$_{min}$ | 6.8 | 6.2 | 16.5 | 3.3 | 0.9 | 272 | 37 | 8.6 | 0.7 | 6.6 | 1.4 | 0.1 | 0.1 | 9.1 | 1.6 | 0.1 |
| B$_{max}$ | 7.6 | 6.9 | 50.5 | 10.1 | 5.8 | 440 | 131 | 18.9 | 5.2 | 15.2 | 6.1 | 0.1 | 0.6 | 25.0 | 9.4 | 1.3 |
| B$_{mean}$ | 7.1 | 6.5 | 30.8 | 6.2 | 2.6 | 311 | 83 | 13.2 | 1.3 | 10.7 | 3.6 | 0.1 | 0.2 | 16.0 | 3.7 | 0.4 |
| SD | 0.3 | 0.2 | 11.7 | 2.3 | 1.5 | 55 | 33 | 4.0 | 1.6 | 2.9 | 1.7 | 0.0 | 0.2 | 4.9 | 2.4 | 0.4 |
| CA | 4.9 | 4.2 | 63.2 | 12.6 | 11.0 | 112 | 2 | 5.7 | 0.4 | 0.3 | 0.2 | 1.7 | 0.3 | 2.9 | 1.5 | 2.0 |
| SD | 0.1 | 0.2 | 2.2 | 0.4 | 1.4 | 10 | 0 | 1.6 | 0.1 | 0.1 | 0.1 | 0.2 | 0.1 | 0.6 | 1.0 | 1.0 |
| CB | 4.8 | 3.8 | 48.5 | 9.7 | 3.8 | 137 | 9 | 12.8 | 0.3 | 1.8 | 0.8 | 2.9 | 0.2 | 6 | 2.3 | 0.4 |
| SD | 0.2 | 0.2 | 2.0 | 0.4 | 0.4 | 10 | 0 | 5.0 | 0.1 | 0.1 | 0.1 | 0.1 | 0.1 | 0.5 | 1.0 | 1.0 |
| | | | | | | 7th sampling—October (Year 3) | | | | | | | | | | |
| A$_{min}$ | 6.7 | 5.9 | 22.0 | 4.4 | 1.8 | 192 | 35 | 6.6 | 0.5 | 8.1 | 1.1 | 0.1 | 0.1 | 11.0 | 3.4 | 0.2 |
| A$_{max}$ | 7.6 | 6.8 | 31.5 | 6.3 | 3.9 | 554 | 65 | 16.3 | 1.4 | 15.7 | 2.4 | 0.1 | 0.2 | 17.5 | 14.0 | 0.8 |
| A$_{mean}$ | 7.2 | 6.4 | 26.1 | 5.2 | 2.6 | 290 | 53 | 11.0 | 0.7 | 11.4 | 1.6 | 0.1 | 0.1 | 14.0 | 7.8 | 0.5 |
| SD | 0.3 | 0.3 | 3.9 | 0.8 | 1.0 | 134 | 12 | 3.6 | 0.3 | 3.0 | 0.5 | 0.0 | 0.0 | 2.7 | 3.9 | 0.2 |

**Table 3.** *Cont.*

| Soil | pH$_{(H_2O)}$ | pH$_{(KCl)}$ | Org C | OM | N | K | P | C/N | K$^+$ | Ca$^{2+}$ | Mg$^{2+}$ | Al$^{3+}$ | Na$^+$ | eCEC | Ca/Mg | K/Mg |
|---|---|---|---|---|---|---|---|---|---|---|---|---|---|---|---|---|
| | | | g kg$^{-1}$ | % | g kg$^{-1}$ | mg kg$^{-1}$ | | | | | | cmol$_{(+)}$ kg$^{-1}$ | | | | |
| B$_{min}$ | 7.0 | 6.2 | 16.5 | 3.3 | 1.8 | 256 | 50 | 9.4 | 0.6 | 7.8 | 2.4 | 0.1 | 0.1 | 11.1 | 1.7 | 0.1 |
| B$_{max}$ | 7.8 | 6.8 | 47.0 | 9.4 | 3.9 | 394 | 109 | 14.9 | 1.0 | 11.7 | 6.4 | 0.1 | 0.2 | 19.2 | 3.2 | 0.3 |
| B$_{mean}$ | 7.4 | 6.4 | 34.5 | 6.9 | 2.9 | 317 | 67 | 12.0 | 0.8 | 9.9 | 4.3 | 0.1 | 0.1 | 15.2 | 2.5 | 0.2 |
| SD | 0.3 | 0.2 | 10.3 | 2.1 | 0.8 | 46 | 20 | 2.3 | 0.1 | 1.5 | 1.5 | 0.0 | 0.0 | 3.0 | 0.6 | 0.1 |
| CA | 4.8 | 4.3 | 58.2 | 11.6 | 11.5 | 112 | 2 | 5.1 | 0.4 | 0.2 | 0.3 | 1.6 | 0.2 | 2.7 | 0.7 | 1.3 |
| SD | 0.1 | 0.2 | 2.3 | 0.5 | 1.1 | 11 | 0 | 2.1 | 0.1 | 0.1 | 0.1 | 0.2 | 0.1 | 0.6 | 1.0 | 1.0 |
| CB | 4.7 | 3.8 | 46.4 | 9.3 | 3.7 | 130 | 8 | 12.5 | 0.2 | 1.8 | 0.7 | 2.6 | 0.2 | 5.5 | 2.6 | 0.3 |
| SD | 0.1 | 0.1 | 1.8 | 0.4 | 0.5 | 9 | 0 | 3.6 | 0.1 | 0.2 | 0.1 | 0.1 | 0.1 | 0.6 | 2.0 | 1.0 |
| | | | | | | | 8th sampling—March (Year 3) | | | | | | | | | |
| A$_{min}$ | 6.4 | 5.7 | 20.5 | 4.1 | 1.4 | 160 | 37 | 9.0 | 0.4 | 6.8 | 1.1 | 0.1 | 0.1 | 9.5 | 3.6 | 0.3 |
| A$_{max}$ | 7.3 | 6.7 | 34.0 | 6.8 | 2.8 | 314 | 52 | 14.6 | 0.8 | 13.7 | 2.6 | 0.1 | 0.2 | 15.6 | 12.5 | 0.5 |
| A$_{mean}$ | 6.8 | 6.1 | 27.0 | 5.4 | 2.3 | 233 | 48 | 11.9 | 0.6 | 10.3 | 1.8 | 0.1 | 0.2 | 12.9 | 6.6 | 0.3 |
| SD | 0.4 | 0.4 | 5.2 | 1.0 | 0.5 | 56 | 6 | 2.3 | 0.1 | 2.5 | 0.6 | 0.0 | 0.0 | 2.1 | 3.6 | 0.1 |
| B$_{min}$ | 6.4 | 5.7 | 17.0 | 3.4 | 1.4 | 160 | 28 | 10.7 | 0.4 | 7.9 | 2.3 | 0.1 | 0.1 | 10.8 | 1.6 | 0.1 |
| B$_{max}$ | 7.3 | 6.6 | 51.5 | 10.3 | 2.8 | 276 | 97 | 21.0 | 0.7 | 10.7 | 5.3 | 0.1 | 0.2 | 15.5 | 4.3 | 0.2 |
| B$_{mean}$ | 6.8 | 6.1 | 32.0 | 6.4 | 2.1 | 218 | 61 | 15.4 | 0.5 | 9.4 | 3.4 | 0.1 | 0.2 | 13.6 | 2.9 | 0.2 |
| SD | 0.3 | 0.3 | 10.3 | 2.1 | 0.5 | 41 | 19 | 3.3 | 0.1 | 1.0 | 1.0 | 0.0 | 0.0 | 1.4 | 0.8 | 0.0 |
| CA | 4.9 | 4.4 | 61.8 | 12.4 | 11.4 | 111 | 3 | 5.4 | 0.3 | 0.3 | 0.3 | 1.7 | 0.2 | 2.8 | 1.0 | 1.0 |
| SD | 0.2 | 0.1 | 2.1 | 0.4 | 1.5 | 11 | 0 | 1.4 | 0.1 | 0.1 | 0.1 | 0.2 | 0.1 | 0.6 | 1.0 | 1.0 |
| CB | 4.9 | 4.0 | 50.2 | 10.0 | 3.8 | 133 | 9 | 13.2 | 0.3 | 1.9 | 0.8 | 2.9 | 0.3 | 6.2 | 2.4 | 0.4 |
| SD | 0.1 | 0.2 | 2.0 | 0.4 | 0.4 | 8 | 0 | 5.0 | 0.1 | 0.1 | 0.2 | 0.1 | 0.1 | 0.6 | 0.5 | 0.5 |
| | | | | | | | 9th sampling—(Year 3) | | | | | | | | | |
| A$_{min}$ | 6.3 | 5.4 | 22.5 | 4.5 | 2.3 | 236 | 48 | 9.9 | 0.6 | 7.4 | 1.2 | 0.1 | 0.2 | 10.2 | 3.8 | 0.3 |
| A$_{max}$ | 7.1 | 6.6 | 33.0 | 6.6 | 3.2 | 536 | 113 | 14.5 | 1.3 | 15.1 | 2.1 | 0.3 | 0.3 | 18.0 | 9.7 | 0.8 |
| A$_{mean}$ | 6.8 | 6.1 | 28.1 | 5.6 | 2.5 | 386 | 70 | 11.5 | 1.0 | 10.3 | 1.7 | 0.1 | 0.2 | 13.3 | 6.5 | 0.6 |
| SD | 0.3 | 0.4 | 4.6 | 0.9 | 0.4 | 97 | 25 | 1.8 | 0.2 | 3.2 | 0.4 | 0.1 | 0.0 | 3.0 | 2.9 | 0.2 |
| B$_{min}$ | 6.4 | 5.5 | 15.5 | 3.1 | 1.4 | 166 | 39 | 8.3 | 0.4 | 5.2 | 2.1 | 0.1 | 0.2 | 8.0 | 2.0 | 0.1 |
| B$_{max}$ | 7.3 | 6.6 | 40.0 | 8.0 | 4.0 | 532 | 127 | 15.4 | 1.3 | 12.4 | 6.3 | 0.1 | 0.3 | 19.9 | 3.2 | 0.5 |
| B$_{mean}$ | 6.8 | 6.0 | 28.0 | 5.6 | 2.6 | 326 | 85 | 11.3 | 0.8 | 8.2 | 3.4 | 0.1 | 0.2 | 12.8 | 2.5 | 0.3 |
| SD | 0.3 | 0.3 | 8.6 | 1.7 | 1.0 | 114 | 32 | 2.4 | 0.3 | 2.3 | 1.3 | 0.0 | 0.0 | 3.6 | 0.4 | 0.1 |
| CA | 5.0 | 4.3 | 61.4 | 12.3 | 11.3 | 110 | 3 | 5.4 | 0.4 | 0.2 | 0.2 | 1.6 | 0.2 | 2.6 | 1.0 | 2.0 |
| SD | 0.2 | 0.3 | 2.3 | 0.5 | 1.3 | 13 | 0 | 1.8 | 0.1 | 0.1 | 0.1 | 0.2 | 0.1 | 0.6 | 1.0 | 1.0 |
| CB | 4.9 | 3.8 | 49.5 | 9.9 | 3.9 | 132 | 9 | 12.7 | 0.3 | 2.0 | 0.8 | 2.8 | 0.2 | 6.1 | 2.5 | 0.4 |
| SD | 0.1 | 0.1 | 2.5 | 0.5 | 0.3 | 9 | 0 | 8.3 | 0.2 | 0.2 | 0.1 | 0.1 | 0.1 | 0.7 | 2.0 | 2.0 |

Org C: organic carbon content; OM: organic matter; eCEC: effective cation exchange capacity; SD: standard deviation; min: minimum value; max: maximum value; mean: average value.

### 3.3. Analysis of the Limiting Factors for Vine Development: Cu and Zn Contents in Soils

The total and available Cu and Zn contents in soils are shown in Table 4 (full data in Supplementary Materials Table S7). The limiting factors for vine development related to Cu and Zn were evaluated over three years (one sampling per year) to establish the total and available contents over time in the two vineyards and their comparison with the respective controls.

- "Cu" factor. None of the vineyard soils analyzed had available Cu deficits (>1 mg kg$^{-1}$; they are not classified as Cu1). All vineyard soils have high available Cu contents, which increased over the 3 years of study. Similar results are found for the total Cu contents, whose highest contents were determined in the third year of sampling. These values are in line with the intensive treatments with fungicides and other phytosanitary compounds to which the vines were subjected during the period studied (Supplementary Materials Table S2). According to the parameters established by the Cu factor, all vineyard soils in plot A in the third year of sampling exceeded 25 mg kg$^{-1}$

of available Cu (classified as Cu2), which is established as phytotoxic for vineyard soils. Likewise, the average available Cu contents in the soils of plot B exceeded 25 mg kg$^{-1}$ since the first sampling was carried out, so the toxicity problems for the vines began before the first sampling. Likewise, total Cu contents in all vineyard soils are above the limit of 100 mg kg$^{-1}$ (classified as Cu3), which is established as phytotoxic [5]. In contrast, none of the control soils exceeded the limits set to be considered as having toxicity problems; this confirms that the origin of the high levels of Cu in the vineyard soils is related mainly to the phytosanitary treatments carried out in the vineyards. In addition, an increase in total and available Cu content was observed in control soils after the third sampling, which may be related to the transfer and mobilization of Cu to areas adjacent to the vineyard.

- "Zn" factor. Available Zn contents in vineyard soils increased during the 3 years of study. During the first two years, Zn contents in the soils were deficient (classified as Zn1 < 3.3 mg kg$^{-1}$). On the other hand, in the third year, the available Zn contents were higher than 15 mg kg$^{-1}$, thus exceeding the limit established to consider the concentrations as phytotoxic (classified as Zn2). In general, total Zn contents also increased during the 3 years in the vineyard soils, but the values are below the phytotoxic value [5]. The increase of Zn content in the soils is mainly associated with the phytosanitary treatments carried out in the vineyards. In contrast, the control soils were deficient in Zn during the entire study period. An increase in total and available Zn content was observed over time in control soils, which may be linked to Zn transfer and mobilization from the vineyard soils to adjacent areas.

**Table 4.** Total and available contents of copper and zinc in vineyard soils and controls.

| Soil (mg kg$^{-1}$) | 1st Sampling (Year 1) | | | | 2nd Sampling (Year 2) | | | | 3rd Sampling (Year 3) | | | |
|---|---|---|---|---|---|---|---|---|---|---|---|---|
| | CuT | CuA | ZnT | ZnA | CuT | CuA | ZnT | ZnA | CuT | CuA | ZnT | ZnA |
| **A$_{min}$** | 206 | 14.6 | 103 | 2.6 | 189 | 15.4 | 115 | 6.5 | 241 | 26.9 | 140 | 16.7 |
| **A$_{max}$** | 210 | 18.2 | 111 | 2.9 | 219 | 20.1 | 142 | 6.8 | 271 | 31.6 | 171 | 18.5 |
| **A$_{(mean)}$** | 208 | 16.1 | 107 | 2.8 | 207 | 17.1 | 127 | 6.7 | 259 | 28.6 | 155 | 17.6 |
| **SD** | 2 | 1.4 | 4 | 0.1 | 12 | 1.7 | 9 | 0.1 | 12 | 1.7 | 12 | 0.7 |
| **B$_{min}$** | 114 | 17.6 | 68 | 1.4 | 123 | 18.2 | 84 | 5.3 | 175 | 55.7 | 130 | 17.9 |
| **B$_{max}$** | 309 | 56.9 | 110 | 2.0 | 324 | 63.7 | 125 | 5.9 | 376 | 92.4 | 167 | 23.1 |
| **B$_{(mean)}$** | 196 | 35.3 | 87 | 1.8 | 207 | 41.3 | 107 | 5.7 | 259 | 76.4 | 143 | 19.8 |
| **SD** | 91 | 15.4 | 16 | 0.2 | 95 | 16.1 | 16 | 0.2 | 95 | 12.9 | 14 | 2.1 |
| **CA** | 39 | 1.2 | 63 | 0.8 | 41 | 1.2 | 67 | 1.5 | 44 | 7.1 | 78 | 2.8 |
| **SD** | 3 | 0.2 | 4 | 0.1 | 3 | 0.1 | 5 | 0.2 | 4 | 1.1 | 3 | 0.4 |
| **CB** | 28 | 1.1 | 61 | 0.9 | 30 | 1.1 | 67 | 1.9 | 53 | 9.7 | 75 | 3.6 |
| **SD** | 2 | 0.2 | 5 | 0.2 | 2 | 0.2 | 7 | 0.3 | 3 | 1.4 | 3 | 0.5 |

### *3.4. Analysis of the Nutritional Status of Vine Leaves*

The contents of the macro- and micronutrients in the petiole and leaf blades collected in the vineyards (plot A and Plot B) for the second and third years of the study are shown in Table 5.

The nutritional analysis of the vine through leaf diagnosis of petioles and leaf blades indicated an important deficiency of P and Ca in both plots, which worsens during the third-year analysis. A slight Zn deficiency was also observed during the third-year analysis. In contrast, the levels of N and K in the leaf blades indicate an excess of both macronutrients, although the N content in petioles was optimal. Excessive Mg contents were also observed during the second-year analysis, although Mg contents decreased in the third-year analysis of the study. The K contents are also excessive in both plots in the second year, although in the third year, they are slightly excessive. Due to the excess of K and Mg in the leaf blades, evaluating the ratio of both nutrients in the petioles is necessary to establish possible deficiencies due to antagonism between nutrients. The results indicated that the K/Mg

ratio in leaf petioles of vine leaves of plot B is optimal. In contrast, in plot A, there is a possible Mg deficiency (despite their high contents in the blade leaf) due to the high K contents found in the leaf petioles. The Fe and Mn contents in the leaf blades are optimal, as well as those of Cu, which, despite exhaustive treatments with Cu-rich fungicides, did not affect the contents in vine leaves.

**Table 5.** Contents of macro- and micronutrients in the leaf blades and petioles.

| | | | | | | | | | | |
|---|---|---|---|---|---|---|---|---|---|---|
| **Sampling—Second Year** | | | | | | | | | | |
| Vineyard/Nutrient ($mg\ kg^{-1}$) | N | | P | | K | | Cu | | Zn | |
| | Lb | Pt | Lb | Pt | Lb | Pt | Lb | Pt | Lb | Pt |
| Plot A | 32.6 | 12.8 | 1.6 | 1.5 | 16.7 | 36.2 | 15.0 | 6.5 | 32.9 | 33.9 |
| SD | 7.8 | 2.5 | 0.8 | 0.4 | 3.1 | 20.1 | 1.4 | 1.0 | 7.6 | 2.4 |
| Plot B | 32.6 | 16.8 | 1.3 | 1.6 | 18.0 | 34.8 | 15.2 | 9.3 | 26.5 | 50.6 |
| SD | 2.3 | 0.6 | 0.4 | 0.8 | 0.3 | 2.2 | 5.4 | 2.9 | 4.1 | 12.3 |
| Vineyard/Nutrient ($mg\ kg^{-1}$) | Mg | | Ca | | Fe | | Mn | | K/Mg | |
| | Lb | Pt | Lb | Pt | Lb | Pt | Lb | Pt | Lb | Pt |
| Plot A | 3.4 | 2.7 | 18.3 | 12.9 | 63.1 | 16.8 | 141.0 | 57.9 | 4.9 | 13.4 |
| SD | 1.5 | 0.7 | 3.8 | 5.1 | 8.8 | 0.7 | 24.6 | 18.9 | - | - |
| Plot B | 3.8 | 3.6 | 11.5 | 11.8 | 61.5 | 16.8 | 108.6 | 82.7 | 4.7 | 9.7 |
| SD | 0.4 | 0.4 | 0.3 | 0.8 | 1.5 | 0.3 | 13.9 | 13.9 | - | - |
| **Sampling—Third year** | | | | | | | | | | |
| Vineyard/Nutrient ($mg\ kg^{-1}$) | N | | P | | K | | Cu | | Zn | |
| | Lb | Pt | Lb | Pt | Lb | Pt | Lb | Pt | Lb | Pt |
| Plot A | 28.0 | 10.7 | 0.9 | 0.7 | 14.4 | 30.1 | 11.0 | 5.3 | 29.3 | 32.5 |
| SD | 7.9 | 3.2 | 0.3 | 0.3 | 3.4 | 14.3 | 1.4 | 1.1 | 8.8 | 2.8 |
| Plot B | 29.4 | 12.6 | 1.0 | 1.1 | 14.6 | 28.1 | 12.3 | 8.0 | 22.3 | 50.3 |
| SD | 2.0 | 2.0 | 0.1 | 0.1 | 0.6 | 1.6 | 3.9 | 2.1 | 2.5 | 10.3 |
| Vineyard/Nutrient ($mg\ kg^{-1}$) | Mg | | Ca | | Fe | | Mn | | K/Mg | |
| | Lb | Pt | Lb | Pt | Lb | Pt | Lb | Pt | Lb | Pt |
| Plot A | 2.5 | 1.9 | 14.8 | 12.2 | 60.3 | 15.8 | 131.9 | 48.1 | 5.8 | 15.8 |
| SD | 0.8 | 0.3 | 4.5 | 4.8 | 9.8 | 0.4 | 22.4 | 20.8 | - | - |
| Plot B | 2.2 | 2.8 | 9.6 | 10.0 | 61.7 | 15.6 | 90.8 | 84.4 | 6.6 | 10.0 |
| SD | 0.2 | 0.1 | 0.1 | 1.1 | 3.1 | 0.1 | 27.1 | 29.9 | - | - |

Lb: leaf blades; Pt: petiole; SD: standard deviation.

## 4. Discussion

From the point of view of soil physical characteristics, a vineyard soil must provide a suitable environment for the growth of the vine. The plants should easily develop their root system, have good aeration, appropriate water retention capacity, and a water circulation regime that does not cause excessive washing or flooding to avoid root asphyxia of the vines [56,57]. The vineyard soils studied have a sandy loam texture, which implies medium permeability, medium-specific surface area, medium compactness, medium nutrient storage capacity, medium water storage capacity, and medium water retention energy. Generally, the stoniness of these soils (including soil controls) is average. It varies between 10.2% in the control for plot B where deciduous vegetation develops and 63.9% in the vineyard soil sample in plot B where a small plateau is formed, and the stones move down the slope from the upper zone of the vineyard deposited. According to stoniness, "E" factor, and "M" factor, the plot B vineyard may have a medium/slight limitation associated with mechanization, trafficability, and easy tillage for the crop, while there are no such limitations for plot A. It was reported that soils with less than 5% stone by volume and with

clay loam to light clay texture have a high potential for vineyard growth, while vineyard soils with sandy to sandy loam texture have a lower potential [58]. The bulk density of the soils varied between 0.96 g cm$^{-3}$ y 1.35 g cm$^{-3}$ in plot A and between 0.98 g cm$^{-3}$ and 1.14 g cm$^{-3}$ in plot B. It was reported that ideal bulk density values in vineyard soils should be less than 1.40 g cm$^{-3}$, so all soils studied were below this critical value [59].

From the point of view of soil chemical characteristics, the pH analysis indicated that the vineyard soils at the beginning of the study were weakly acidic. With the addition of compost (with basic pH) to the vineyard soils for two years, the pH of these soils increased to *neutral* values. Different authors have indicated that the pH of vineyard soils can be from acidic (pH < 5.5) to basic soils (pH 8.5), although in terms of vine cultivation, a neutral pH is preferable [60–64]. In our study, the treatment with compost had a positive effect on the soils of both plots, increasing the pH and favoring the maintenance of the soil pH at neutral values, so its regular use is recommended for this slightly acidic soil.

As mentioned above, the soils of both plots have a sandy loam texture, which implies that they have a high percentage of sand. Thus, the soils must have a good level of organic matter to replace the lack of mineral colloids and thus achieve correct water retention and cation exchange capacities. The level of organic matter for sandy loam texture soil may be around 2.2%. Generally, soils with less than 10% clay should be at least between 1.4% and 1.8% [65]. All the vineyard soil samples showed organic matter percentages between 1.4% and 3.1% in plot A and between 2.6% and 4.3% in plot B before adding the compost (first sampling). Thus, some soils in plot A are below the optimum values and can be considered as having some organic matter deficit (classified as om1). Subsequently, with the application of compost, the average organic matter contents in all vineyard soils increased above 5%, which could be considered an excess for vineyard soils (classified as om2). However, the control soils have organic matter percentages above 5% and are higher than the vineyard soils, even after compost application. This indicates that the adjacent control forest soils, which had not undergone management processes, have much higher organic matter contents. This fact corresponds to the high natural richness in organic material of the soils from Galicia [66]. This confirms that the management practices to which both plots are subjected suffered a very high loss of organic material. This makes it necessary to have an external contribution of organic material to maintain adequate levels to avoid reducing the organic matter content and favoring soil structuring, limiting the risk of erosion, and increasing the water storage capacity [67]. Therefore, although the organic matter contents are above the general values recommended in the literature for vineyard soils, these values are more in line with the typical values of Galician soils, specifically in the Rías Baixas D.O. [66].

Regarding the N content in the vineyard soils, in general, they presented normal values, except just after compost application. The high N contents in the vineyard soils may be related to the excess N observed in the foliar analysis. Likewise, high N contents in vineyard soils may provoke an excess of plant vigor and favor excessive plant growth. This leads to a microclimate deterioration of the leaves and clusters, affecting the setting and ripening processes of the grapes and their quality and promoting the development of grapevine diseases, which is necessary to increase phytosanitary treatments [68]. In addition, due to the high organic C contents in soils, the C/N ratios were elevated in all vineyard soils. During the first sampling, C/N ranged from 11 to 18 in plot A, while in plot B, it ranged from 38 to 108. After compost application, the C/N ratio decreased to values that ranged from 6.6 to 18.9. Despite this, C/N >10 indicates that the soils have an excess of carbon.

The available phosphorus contents of vineyard soils ranged between high (36–72 mg kg$^{-1}$) and very high (high > 72 mg kg$^{-1}$) according to the *"P" factor*. On the other hand, the control soils showed phosphorus deficiency. (<18 mg kg$^{-1}$). The high and very high P contents in the vineyard soils are a consequence of applying both composts (rich in P assimilable) and fertilization with high phosphorus contents in the vineyard in the years before this study. The low assimilable P contents in the control soils are associated

with the severe P deficiency suffered by natural soils in Galicia, which is a major constraint for agricultural production [69]. This low P availability is usually associated with soils with acidic pH due to the precipitation of P in the form of Fe or Al phosphates and its adsorption by soil colloids [70,71]. Normally, the correction of P deficiency in soils is linked to applying phosphate fertilizers or compost with high organic C content, which brings P assimilable to the soil. However, the application of these compounds is not always reflected in the capacity of the vines to assimilate this element since it increases the potential for P available transfer through the soil profile, which represents a greater potential for surface and groundwater contamination [72–74]. The high contents of assimilable P measured in the vineyard soils of both plots were not reflected in the P counts in the vine leaves, which showed a significant P deficiency.

The K contents in the soils are relatively high. These high K contents can favor crop resistance to cold and periods of drought and plant structure and stiffness; K also increases resistance to pests. This macronutrient exerts a very positive effect on the quality of the grapes, balancing the unfavorable effects of N excess. On the other hand, a certain imbalance between Ca and Mg content (ratio > 10) was observed in some vineyard soils, as well as between K and Mg (ratio > 0.5) in the two plots. These imbalances directly may affect the vines according to several studies [75,76] since the nutritional analysis through foliar diagnostics indicated that there is an excess of K. This causes an imbalance with elements such as Ca, which is deficient in the studies vines, and Mg, which although it is not deficient, its imbalance with K makes the nutritional values not optimal in the vines studied. This suggests that it is necessary to continue with the addition of magnesian limestone (Supplementary Materials Table S2) to the vineyard soils, at least until their contents are balanced with the K. In addition, none of the vineyard soils showed limitations associated with eCEC after compost application. Thus, regulation of the exchange cations supplied to the vineyard soils may be the solution to improve cationic nutrient imbalances.

The available and total Cu contents exceeded the limits established to be considered phytotoxic (Section 2.4) [5]. Several studies have found high Cu contents in vineyard soils due to the intensive and excessive use of Cu-based fungicides (Supplementary Materials Table S2) [77–79] and that these soils need remediation strategies since high Cu contents generate a high risk of toxicity in young vine plants and in subsequent crops replacing vines [12,14]. The foliar diagnosis confirms that the Cu contents in the blade leaves are within the values considered optimal, which despite the high contents in the soil, do not pose a risk to the development of the vines in both plots. Even so, an increase in Cu content was observed in the control soils where no phytosanitary treatment was carried out. This indicates that, although slowly, there is a possible transfer of Cu from the vineyard soils to adjacent areas, which in the medium and long term, may cause toxicity problems in other species more sensitive to this metal.

The available Zn contents in the vineyard soils were deficient during the first two years of sampling, but in the third year, the values reached phytotoxic values. According to the treatments applied to the soils (Supplementary Materials Table S2), the continuous treatment of the vines with compounds such as Zineb, Mancozeb, Cimox, and Ethylene-bisdithiocarbamate favored the accumulation of Zn in the vineyard soils. In contrast, the control soils are deficient in Zn. The leaf diagnosis in vines shows that despite the treatments applied, the Zn contents in the leaf blades showed that the vines have a slight Zn deficiency, mainly during the third-year analysis. This may be due to the high P contents in the soils since several studies have pointed out that excess P in soil can block Zn absorption, causing chlorosis problems and reduced vigor and growth of the vines [80].

## 5. Conclusions

This study highlighted the importance of identifying and evaluating the soil factors that can limit vineyard production within the Rías Baixas appellation and thus establish strategies for improving management based on soil characteristics, phytosanitary treatments applied, and fertilization and tillage practices. The soil physical factors showed that

they can cause some limitations associated with mechanization, trafficability, and ease of tillage for the crop, mainly in the plot with the steepest slope. The sandy loam soil texture is not ideal. Still, the application of compost in the vineyard improves soil conditions and replaces the lack of mineral colloids, enhancing water retention and cation exchange capacity. The evaluation of soil chemistry factors confirmed that the weakly acidic character of the vineyard soils must be corrected with the appropriate application of compost to favor neutrality. The high N content in some soils and the excess in vines can lead to excessive vigor and an excessive increase in plant growth, with consequences on grape quality and an increase in cryptogamic diseases. The high K content in grapevines compensates for excess N, but its control is necessary to avoid nutrient imbalances with other cations such as Ca and Mg. Despite the high available phosphorus contents in the vineyard soils, the deficiencies in vine leaves are a symptom that the application of phosphate fertilizers and compost to the soils has not been an effective measure to increase the available P content in these soils. Due to phytosanitary treatments, total and available Cu contents were above the phytotoxic limits in the soils. The available Zn content also exceeded phytotoxic limits after the continuous application of phytosanitary treatments. The Cu contents in the grapevines do not pose a risk for the development of the species, while the vines showed Zn deficiencies, which may be linked to the high P contents that block its absorption. Therefore, it is necessary to regulate and limit phytosanitary treatments as much as possible to avoid the accumulation or transfer of Cu and Zn metals to other parts of the ecosystem. In conclusion, the soil factors evaluated are adequate to know and apply the necessary corrective measures to know the needs of vineyard soils to have quality plant production.

**Supplementary Materials:** The following supporting information can be downloaded at: https://www.mdpi.com/article/10.3390/agronomy13020309/s1.

**Author Contributions:** Conceptualization, C.C.-C. and D.A.-L.; Methodology, R.V.-B.; Formal analysis, R.V.-B., R.G.-F. and C.C.-C.; Investigation, R.G.-F., C.C.-C. and D.A.-L.; Writing—original draft, R.V.-B. and R.G.-F.; Writing—review and editing, D.F.-C. and D.A.-L.; Funding acquisition, D.F.-C. All authors have read and agreed to the published version of the manuscript.

**Funding:** This work was funded by the COPPEREPLACE project, which has received funding from (grant agreement SOE4/P1/E1000) FEDER funds through the INTERREG- SUDOE program, and by Xunta de Galicia via the BV1 research group (ED431C 2017/62-GRC).

**Institutional Review Board Statement:** Not applicable.

**Informed Consent Statement:** Not applicable.

**Data Availability Statement:** All data are show in the manuscript and Supplementary Materials.

**Acknowledgments:** David Fernández-Calviño holds a Ramón y Cajal contract (RYC-2016-20411) financed by the Spanish Ministry of Economy Industry and Competitiveness. Daniel Arenas-Lago thanks the Ministerio de Ciencia e Innovación of Spain and the University of Vigo for the postdoc grant Juan de la Cierva Incorporación 2019 (IJC2019-042235-I). Claudia Campillo-Cora holds a predoctoral fellowship with Xunta de Galicia (ED401A-2020/084) funded by Consellería de Educación, Universidade e Formación Profesional. We want to thank Santiago de Laiglesia Gil and Rocío Bravo Xicotencatl, who are responsible for the Bodega Gil Armada and the vineyards studied, for their great help in the different samplings, for providing us with detailed information on the treatments carried out throughout the study period, and for their total availability to collaborate with the authors of this work.

**Conflicts of Interest:** The authors declare no conflict of interest.

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
