# Peer review of "Risk Assessment and Limiting Soil Factors for Vine Production—Cu and Zn Contents in Vineyard Soils in Galicia (Rías Baixas D.O.)"

_agronomy, doi:10.3390/agronomy13020309_

Round 1

Reviewer 1 Report

1.      Why the soil sample is 25-30 cm?

2.      The contnet of OM and OC could only show one of them.

Author Response

  1. Why the soil sample is 25-30 cm?

Thank you very much for the comment. We have corrected the sentence and clarified the depth at which the samples were taken: "Soil sampling was carried out in the two plots (A and B) taking samples of the upper layer of soil (0-30 cm) at different points".

The soil samples from both vineyards were taken from the first 30 centimeters, which is the effective depth of the soils of the plots that coincides with the superficial layer of the soils of the vineyards.

  1. The content of OM and OC could only show one of them.

Thank you very much for your suggestion. Although carbon content and percentage of organic matter are related (as we have now included in the text), we believe that we should not eliminate either for the following reasons:

(a) We consider that the organic matter content in necessary to show because from that value is determined the "om" factor, which is used to evaluate the organic matter content. It is considered defi-cit contents < 2% (om1) and excess contents > 5% (om2).

(b) We consider that it is important to keep the carbon content in the manuscript because from this value, together with that of N, the C/N ratio is calculated, which is also indicated in the table of soil characteristics.

Reviewer 2 Report

This study analyse two areas of vineyard soils in Rias Baixas (Galicia).

The objectives are clear, but I suggest it can be improved. The study analysed the soil factors during three years. In this three years a organic amendment was applied so it must be described in the objetives.

Table 3 and 4 can be changed by graphs, this can improve the understanding of the paper.

Author Response

This study analyse two areas of vineyard soils in Rias Baixas (Galicia).

The objectives are clear, but I suggest it can be improved. The study analysed the soil factors during three years. In this three years a organic amendment was applied so it must be described in the objetives.

Thank you very much for your suggestion. We have added your recommendation to the objectives.

Table 3 and 4 can be changed by graphs, this can improve the understanding of the paper.

Thank you very much for your suggestion, but finally we have decided to use the tables because we consider that they are too much data to put in graphs. In addition, with the data provided in the supplementary material, we believe that the results as a whole are clearer if the numerical data are provided. In addition, the values of the soil characteristics in most of the scientific articles are given in tables.

Reviewer 3 Report

The manuscript entitled « Risk assessment and limiting soil factors for vine production Cu and Zn contents in vineyard soils in Galicia (Rías Baixas D.O.).» is scientifically exciting and original.

 However, there are some points in the manuscript that need improvement:

 1.     In my opinion, the words in the paper's title should not be repeated in the keywords. So the words “soil factors; vineyard soil” should be deleted or replaced by other words or phrases.

2.     The text has a long introduction with a rich bibliography, but needs improvement in the English language, from a native English speaker.

3.     The data did not undergo any statistical analysis. When there are results in such important effervescent parameters, it is not enough to find only the average, maximum and minimum values. To decide whether there is a statistically significant difference between the years and among different soils, the ANOVA statistical method needs to be applied. Therefore, the tables with the data cannot be evaluated.

4.     Authors should also perform statistical analysis between the results of different samplings in order to decide whether there is a statistically significant difference between them.

5.     It is necessary to refer to the “materials and methods” chapter, the detection limits of all the scientific instruments used.

6.     Soil properties are largely determined by how the soil was formed, by the minerals and rocks that erode and form the soil layer. There needs to be a discussion regarding nutrient movement and availability related to soil classes or soil taxonomy orders. Since the experiment was carried out on soils of different types, a small paragraph needs to be added to the chapter of the introduction concerning the soil taxonomy orders of the soil used, which would lead to a more holistic approach to the subject. This would be useful for the explanation of the results, too. Soil classes or soil orders are of decisive importance for many of the properties related to the retention or not of nutrients in the solid phase of soils. Regarding the properties and characteristics of soil taxonomy order, information can be used from “Alloway” s and “Kabata-Pendias” Soil Science books. In addition, the following article (among others) might be useful:

 “Pollution assessment of potentially toxic elements in soils of different taxonomy orders in central Greece. Environ Monit Assess (2019) 191: 106, https://doi.org/10.1007/s10661-019-7201-1”

7. IN the discussion section, concerning the effect of organic matter on soil nutrient availability, you could refer to the article: DOI: 10.1007/s11356-020-09165-6, “Chemical and ecotoxicological assessment of sludge-based biosolids used for corn field fertilization”.

Author Response

The manuscript entitled « Risk assessment and limiting soil factors for vine production Cu and Zn contents in vineyard soils in Galicia (Rías Baixas D.O.)» is scientifically exciting and original.

  However, there are some points in the manuscript that need improvement:

  1. In my opinion, the words in the paper's title should not be repeated in the keywords. So the words “soil factors; vineyard soil” should be deleted or replaced by other words or phrases.

Thank you very much for your suggestion. We have deleted both keywords and we have replaced them with “fungicide” and “phytosanitary”.

  1. The text has a long introduction with a rich bibliography, but needs improvement in the English language, from a native English speaker.

Thank you very much for your suggestion. We have sent the document to be reviewed by a native speaker.

  1. The data did not undergo any statistical analysis. When there are results in such important effervescent parameters, it is not enough to find only the average, maximum and minimum values. To decide whether there is a statistically significant difference between the years and among different soils, the ANOVA statistical method needs to be applied. Therefore, the tables with the data cannot be evaluated.

Thank you very much for your suggestion. We have applied the ANOVA statistical analysis to the data and we have added it to the tables into the supplementary data file.

  1. Authors should also perform statistical analysis between the results of different samplingsin order to decide whether there is a statistically significant difference between them.

According the above comment, we have incorporated the ANOVA statistical analysis to the data and the tables into the supplementary data file.

  1. It is necessary to refer to the “materials and methods” chapter, the detection limitsof all the scientific instruments used.

Thank you very much for your suggestion. We have indicated the detection limits in a paragraph in the material and method chapter.

  1. Soil properties are largely determined by how the soil was formed, by the minerals and rocks that erode and form the soil layer. There needs to be a discussion regarding nutrient movement and availability related to soil classes or soil taxonomy orders. Since the experiment was carried out on soils of different types, a small paragraph needs to be added to the chapter of the introduction concerning the soil taxonomy orders of the soil used, which would lead to a more holistic approach to the subject. This would be useful for the explanation of the results, too. Soil classes or soil orders are of decisive importance for many of the properties related to the retention or not of nutrients in the solid phase of soils. Regarding the properties and characteristics of soil taxonomy order, information can be used from “Alloway” s and “Kabata-Pendias” Soil Science books. In addition, the following article (among others) might be useful:

 “Pollution assessment of potentially toxic elements in soils of different taxonomy orders in central Greece. Environ Monit Assess (2019) 191: 106, https://doi.org/10.1007/s10661-019-7201-1”

We know the importance of what you propose, but we believe that insert a paragraph in the introduction section about soil classification does not provide relevant information to our study. On the other hand, we do consider that soil classification is important, so that the readers of the article can know what type of soils were used, we have incorporated the classification of our soils in the material and methods section (2.1. study area).

  1. IN the discussion section, concerning the effect of organic matter on soil nutrient availability, you could refer to the article: DOI: 10.1007/s11356-020-09165-6, “Chemical and ecotoxicological assessment of sludge-based biosolids used for corn field fertilization”.

Thank you very much for your suggestion. We have added the reference to the discussion section.